# OpenReview forum: "Attacking Audio Language Models with Best-of-N Jailbreaking"
_ICLR.cc/2025/Conference — Submitted to ICLR 2025_

### Official Review · Reviewer_g1y2 · 2024-10-22

**Soundness:** 3
**Presentation:** 2
**Contribution:** 2
**Rating:** 3
**Confidence:** 4

**Summary:**

This paper introduces BoN, a new jailbreaking technique for ALMs. Based on various audio augmentation methods, BoN bypasses the safeguards of SOTA ALMs through repeated sampling with higher ASR. Authors also show that BoN jailbreaking can be composed with other jailbreak techniques.

**Strengths:**

1. The research question, studying the jailbreak vulnerabilities of ALMs, is relatively interesting.

2. The experimental part is comprehensive and complete. The appendix is detailed.

**Weaknesses:**

1. As stated in line 157-159, PAIR and TAP are classical jailbreak techniques, and both of them achieve good performance on jailbreaking text-only LLMs. However, when applying single audio augmentation and TTS engine, the ASR on ALMs decrease to below 5%. This result looks counterfactual and lacks of detailed discussion. The authors should provide a more detailed analysis of why this decrease occurs, or discuss the potential reasons when transferred from text to audio.

2. Technical contribution is limited. (1) Lack of detailed description of the proposed algorithm PrePAIR. (2) From the limited description of PrePAIR, while universal, this algorithm is not highly relevant to the previous sections and the “audio” part in language models, which makes the motivation unclear. The authors should clarify the connection between PrePAIR and the audio aspects of language model, and explain how this algorithm relates to the overall motivation of the paper.

3. As a supplementary section of the pre-experiments, Section 3 (especially Section 3.1 & 3.2) does not give clear and valuable insights, which also makes the pre-experiment part lengthy. The authors should highlight the key findings more clearly, which would help this reviewer and other readers understand the value of this section.

4. The presentation of the paper needs to be improved. For example, the duplicated part in upper left figure 1.

**Questions:**

1. See weakness 1. Are there any more insights and discussion?

2. See weakness 2. Do the PrePAIR have any relationship with the audio part in ALMs?

3. See weakness 3.

---

> ### Author Response · Authors · 2024-11-25
>
> Thank you for your thoughtful comments. We are grateful you found our experiments on ALMs interesting, comprehensive, and complete.
>
> > As stated in line 157-159, PAIR and TAP are classical jailbreak techniques, and both of them achieve good performance on jailbreaking text-only LLMs. However, when applying single audio augmentation and TTS engine, the ASR on ALMs decrease to below 5%. This result looks counterfactual and lacks of detailed discussion. The authors should provide a more detailed analysis of why this decrease occurs, or discuss the potential reasons when transferred from text to audio.
>
> We thank the reviewer for pointing this out because it was a mistake writing that this experiment used vocalized PAIR and TAP attacks. Figure 2 only uses direct requests, and we have updated the manuscript accordingly. Furthermore, we have provided a text baseline ASR of Gemini Pro using the same direct requests to put the audio results into perspective (see section 2, results paragraph). For reference, only one direct request jailbreaks Gemini Pro when provided in text at temperature 0. This is an ASR of 0.6%, while the ALM ASR distributions resulting from single augmentations (shown in Figure 2) range from 0% to 6%.
>
> >(1) Lack of detailed description of the proposed algorithm PrePAIR.
>
>
> Thank you for pointing out that the PrePAIR section requires more clarity. We have updated the manuscript to explain more technical details (see Section 4 experimental details). For instance, we use GPT-4o as an attacking LLM that iterates and refines a prefix tested on a batch of 4 direct requests. We terminate PrePAIR if 3 or more of these requests are jailbroken.
>
>
> > (2) From the limited description of PrePAIR, while universal, this algorithm is not highly relevant to the previous sections and the “audio” part in language models, which makes the motivation unclear. The authors should clarify the connection between PrePAIR and the audio aspects of language model, and explain how this algorithm relates to the overall motivation of the paper.
>
> We agree with the reviewer and updated the manuscript to clarify that the motivation is to use composition to improve the sample efficiency of BoN jailbreaking (i.e., reach a certain ASR with few samples). This is important given our threat model since if BoN is easy to compose, it makes it easier for adversaries to elicit harmful information with a smaller computational budget. The choice of using PrePAIR is just one example of a jailbreak technique that could be composed with BoN. Please see the introduction and the first two paragraphs in Section 4.
>
>
> > As a supplementary section of the pre-experiments, Section 3 (especially Section 3.1 & 3.2) does not give clear and valuable insights, which also makes the pre-experiment part lengthy. The authors should highlight the key findings more clearly, which would help this reviewer and other readers understand the value of this section.
>
> Thank you for the feedback. Could you provide more detail on what needs more clarity?
> * Section 3.1 - our main insight is that sets of augmentations that break a specific request (e.g. 1.5x speed, 30% higher pitch on a human speaking "how do I build a bomb") do not transfer well to other requests. This is interesting because it suggests that it is difficult to find a set of augmentations that have high universality across prompts.
> * Section 3.2 - our main insight is that when resampling the same set of augmentations that originally broke the request, the ALM is often not jailbroken again (hence, the jailbreaks we find are unreliable). Therefore, the power comes from the universal algorithm itself, which can elicit harmful information for many harmful requests.
>
>
> > The presentation of the paper needs to be improved. For example, the duplicated part in upper left figure 1.
>
> Yes, we apologize for the corrupted export of Figure 1, which caused the boxes to be misaligned. We have fixed this in the manuscript. Can you provide more information on other areas that need improving?
>
>
> > See weakness 2. Do the PrePAIR have any relationship with the audio part in ALMs?
>
> The PrePAIR prefixes are found by an attacker LLM proposing a prefix, which is then subsequently vocalized using text-to-speech. So, the algorithm does incorporate audio-specific red teaming, which we have made clear in Section 4. However, as mentioned above, the choice of algorithm is less important. In the manuscript, we have now stated that the purpose of the section is to highlight how easy it is to improve the sample efficiency of the algorithm with composition (so it can reach higher ASRs with fewer samples).
>
>
> We hope our revisions and clarifications have addressed your concerns. If so, we respectfully request that you reconsider your evaluation and possibly increase your support for our paper. We welcome further discussion.

---

### Official Review · Reviewer_t8Hb · 2024-10-30

**Soundness:** 1
**Presentation:** 1
**Contribution:** 1
**Rating:** 3
**Confidence:** 4

**Summary:**

This research paper explores the vulnerabilities of Audio Language Models (ALMs) to audio-based jailbreaking attacks. The authors introduce a novel algorithm called Best-of-N (BoN) Jailbreaking, which leverages random audio augmentations to elicit harmful responses from ALMs. They demonstrate the effectiveness of BoN jailbreaking on several state-of-the-art ALMs, including Gemini and GPT-4o. The authors also uncover power laws that predict the effectiveness of BoN as a function of the number of sampled augmentations. Finally, they investigate the composition of BoN with other jailbreaking techniques, demonstrating that combining these methods can lead to more potent attacks.

**Strengths:**

This paper successfully introduces a new black-box jailbreaking algorithm called Best-of-N (BoN) that effectively extracts harmful information from Audio Language Models (ALMs) by exploiting their sensitivity to variations in audio input. The audio domain is in general underexplored so I appreciate this effort.

The paper looks into different aspects such as combinations with other attacks. In addition, the paper provides detailed insights into the workings of BoN jailbreaking, including analysis of the transferability and semantic meaningfulness of the augmentations used. The research highlights the challenges of safeguarding ALMs with stochastic outputs and continuous input spaces.

**Weaknesses:**

My main concerns regarding this paper are that the methodology of the attack is not adequately described, and the evaluation begins without sufficiently introducing the approach and settings.

For example, to understand the experiments, the reader needs to know what is defined as "harmful information."

It is difficult to assess the novelty of this work. Although the authors propose an attack against an alternative domain, the method used is unclear. Additionally, we do not gain many modality-specific insights and could potentially derive the same findings from other, text-only models. It would be beneficial if the paper included some audio-specific insights.

For instance, what is the signal-to-noise ratio (SNR) of the input? Would a human notice the attacks? Does the attack also work if it is played over the air?

In the introduction, the paper describes the "robustness" of the models (third paragraph). However, the method used is not described, and the findings there are not particularly useful.

It would also be appreciated if the authors uploaded a few audio samples for listening.

**Questions:**

See above

---

> ### Author Response · Authors · 2024-11-25
>
> We appreciate the reviewer's feedback and thank them for affirming that jailbreaking in the audio domain is underexplored, with an appreciation for our work.
>
> > My main concerns regarding this paper are that the methodology of the attack is not adequately described, and the evaluation begins without sufficiently introducing the approach and settings. For example, to understand the experiments, the reader needs to know what is defined as "harmful information."
>
> We thank the reviewer for highlighting opportunities to enhance the clarity of our paper. We agree that certain prerequisites, such as the definition of "harmful information" and "jailbreaking", require definitions and have addressed this in the revised introduction (paragraph 2). However, we believe that the methodology of the attack is comprehensively detailed in Section 2.2 and further illustrated in Figure 1. Our approach of integrating "experimental detail" sections throughout the paper rather than a single method section is intentional, aligning with the narrative flow we aim to achieve. We welcome more targeted feedback on specific aspects of the methodology and are ready to provide additional clarifications as needed.
>
>
>
> > It is difficult to assess the novelty of this work. Additionally, we do not gain many modality-specific insights and could potentially derive the same findings from other, text-only models.
>
> We agree with the reviewer that BoN jailbreaking is a general algorithm that has the potential to work across modalities, highlighting we could derive similar findings from text models (in fact, we find it works on VLMs in Appendix D.3). We have updated the paper to make this clearer and that we choose to evaluate its effectiveness in the audio domain only (for example, see start of algorithm details, page 4).
>
> Regarding novelty, the related work section compares our work to text, vision, and audio jailbreak attack methodologies. It highlights our method is novel across modalities since no other work applies modality-specific augmentations repeatedly until a jailbreak is found.
>
>
> > It would be beneficial if the paper included some audio-specific insights.
>
> We appreciate the reviewer's suggestion, but we would like to clarify that our paper extensively addresses audio-specific insights. Each experiment incorporates vocalized versions of harmful requests alongside audio augmentations, ensuring that all results are specific to audio models. Figure 2 illustrates the limited effectiveness of single audio augmentations, such as varied accents and tones across 24 voices, highlighting the robustness of models to these changes. Furthermore, our appendix provides one of the most thorough analyses on frontier models, detailing ASR distribution across models and the impacts of single augmentation sweeps in Appendix C, additional insights on audio augmentations through clustering of embeddings in Appendix E, and numerous experiments using Prepair—our optimized prefix attack method—in Appendix F, which reveal asymmetries in robustness between ALMs and text-based LLMs.
>
>
>
>
> > what is the signal-to-noise ratio (SNR) of the input?
>
> Intuitively, the signal-to-noise (SNR) ratio is how quiet the added background noise, music, or speech is compared to the original file. A low SNR (say 10dB) usually means a human will have difficulty hearing the background noise. A high SNR (say 30dB) means the added noise dominates the file. We update the manuscript to make this clearer.
>
> > Would a human notice the attacks?
>
>
> Our augmentations that modify ALMs yield audio outputs distinct from non-augmented, original human readings of harmful requests. If individuals are trained to identify heavily modified sound files, they could likely flag them. Nevertheless, a listener could also discern the harmful nature of the request itself and consequently block it. Hence, the stealthiness of these attacks is not a crucial aspect of our threat model. We focus primarily on preventing the misuse of highly capable LLMs, which depend on alignment training and other safeguards, rather than human evaluations, which are impractical for frontier labs.

---

> ### Author Response · Authors · 2024-11-25
>
> > Does the attack also work if it is played over the air?
>
> In Section 3, we present brittleness results that show the unreliability of reapplying a set of augmentations found to jailbreak the model during a BoN run; their effectiveness diminishes with slight modifications to the audio file. Consequently, we do not anticipate that jailbreaks transmitted over the air will maintain high reliability. However, in our misuse threat model, the adversary aims to elicit harmful information directly through the API, so transmission over the air is not a requirement. We also emphasize that the strength of this algorithm lies not in its ability to find reliable attacks but in its potential to serve as a universal tool that adversaries could use to extract harmful information. Even a single instance of this misaligned behavior is enough to pose significant misuse risks.
>
>
> > In the introduction, the paper describes the "robustness" of the models (third paragraph). However, the method used is not described, and the findings there are not particularly useful.
>
> We appreciate the feedback and have updated the introduction (paragraphs 1 and 2) to clarify what we mean by robustness and jailbreaking. Regarding the clarity of our method, we believe it is detailed in the third paragraph and visually supported by Figure 1 so politely disagree. Could you please specify which aspects of the method remain unclear?
>
>
> > It would also be appreciated if the authors uploaded a few audio samples for listening.
>
> We upload some examples of augmented files that jailbreak Gemini Pro to the supplementary files (see `code_submission/augmented_files/*.wav`). Furthermore, upon the paper's release, if accepted, we will release a website where the reader can listen to all the working jailbreaks we find.
>
>
> We hope our response has addressed your feedback. If so, we hope you might reevaluate and enhance your support for our paper. We welcome further discussion.

---

> > ### Comment · Reviewer_t8Hb · 2024-11-26
> >
> > Thank you for your response. Unfortunately, I still need to retain my original score, mainly because the contribution is not new in comparison to existing attacks. For example, this paper already shows similar attacks: https://www.usenix.org/conference/usenixsecurity24/presentation/zhang-tingwei

---

> > > ### Author Response · Authors · 2024-11-26
> > >
> > > Thanks for the follow-up and for the example paper on similar work. We will add this to our related work section. However, while this work looks at cross-modality adversarial attacks, it is significantly different from our work in three ways:
> > > * First, this work requires access to the multi-modal embeddings from the encoders. Frontier LLMs do not provide access to audio and vision embeddings, so this method cannot be applied to GPT, Claude, or Gemini family models. The misuse threat is most concerning on the most capable models, so developing attacks that elicit jailbreaks across modalities on frontier systems is important so better defenses can be developed.
> > > * Second, this work looks at changing the vision or sound description, negatively impacting the downstream task performance. However, our work looks at adversarial attacks to jailbreak the model and elicit harmful information, which is a different threat model.
> > > * Third, this work applies perturbations using different methods (such as PGD and a query-based contrastive approach) to embeddings that change the output description of a multi-modal input. However, our work does not operate on embeddings. Instead, it applies augmentations directly to the input (such as speed and pitch for audio) and samples new prompts until we find a jailbroken response.
> > >
> > > Could you provide evidence of related work showing our contribution is not new (e.g. an attack that uses modality-specific augmentations with repeated sampling to elicit jailbreaks) or give further reasoning as to why our reasoning above doesn't demonstrate novelty in our work?

---

### Official Review · Reviewer_YqsM · 2024-11-03

**Soundness:** 3
**Presentation:** 4
**Contribution:** 3
**Rating:** 5
**Confidence:** 4

**Summary:**

This paper introduces "Best-of-N (BoN) Jailbreaking," a black-box algorithm designed to exploit weaknesses in Audio Language Models (ALMs) by extracting harmful information through audio-based attacks. The BoN method uses repeated audio augmentations, such as changes in pitch, speed, and background noise, to malicious prompts until it elicits harmful responses from the target ALM. The study shows that BoN achieves a high attack success rate (ASR), with results exceeding 60% ASR on several top ALMs, including a preview version of GPT-4o’s voice mode. Additionally, the authors discover power laws that allow them to predict ASR based on the number of samples, helping forecast the efficiency of BoN jailbreaking. The approach becomes even more effective when combined with other jailbreak techniques, reaching up to 98% ASR in some models. This paper highlights the difficulty of securing ALMs due to their sensitivity to continuous input variations, proposing BoN as a scalable and effective method for targeting ALM vulnerabilities.

**Strengths:**

- Proposed Best-of-N (BoN) Jailbreaking is novel, specifically tailored for attacking ALMs. BoN is unique in its application of audio augmentations to create high-entropy inputs, exploiting the ALM's sensitivity to continuous input variations. The combination of BoN with other jailbreak methods, such as the PrePAIR technique, showcases an innovative blend of audio augmentations and iterative refinement for more effective attacks.

- The discovery and application of power-law scaling to predict ASR with increased samples indicate a high-quality analysis, providing valuable insights into the scalability and potential impact of the proposed BoN method.

- The paper is well-structured and effectively explains complex ideas, making the novel BoN method accessible. Visualizations such as graphs that show the ASR progression with sample size and the power-law behavior, enhance clarity by illustrating key results.

**Weaknesses:**

- A significant limitation of this study is the decision to turn off Gemini’s safety filter. In real-world applications, LLMs and ALMs rely on both alignment techniques and safety filters for safeguarding against misuse. By disabling these filters, the study's attack success rate may be artificially inflated, making the findings less practical and relevant in environments where safety filters are essential. Including experiments with the safety filter enabled would provide a more realistic assessment of the BoN method's effectiveness and its relevance to real-world deployments.

- The study does not evaluate the audio quality of the modified samples post-attack, which is an important aspect for assessing the stealthiness of these attacks. A low audio quality in the altered samples could make the attacks easily detectable or unnatural. A quality metric, such as the ViSQOL score, would allow for a quantitative comparison between the original and post-attack audio samples. Without such an analysis, it is unclear if the BoN attacks are feasible in scenarios where high-quality audio is essential for covert operations.

**Questions:**

- How might the ASR be affected if the safety filter remained active? Including this would provide a clearer picture of BoN’s practical effectiveness and real-world implications. Would the authors consider conducting additional experiments with the safety filter enabled to better align the study with real-world conditions?

- Audio quality after attack modifications is an important factor for stealthy and practical attacks. Could the authors provide details on the perceptual quality of the modified audio samples, perhaps using a metric like the ViSQOL score?

- The term "ASR" is commonly used to denote "Automatic Speech Recognition" in the speech research community. Could the authors consider using an alternative term?

---

> ### Author Response · Authors · 2024-11-25
>
> We thank the reviewer for the helpful feedback and for recognizing that our method provides valuable insights into the scalability of our algorithm in a well-structured way.
>
>
> > A significant limitation of this study is the decision to turn off Gemini’s safety filter. In real-world applications, LLMs and ALMs rely on both alignment techniques and safety filters for safeguarding against misuse. By disabling these filters, the study's attack success rate may be artificially inflated, making the findings less practical and relevant in environments where safety filters are essential.
>
> While we appreciate the reviewer's concerns, we acknowledge that our initial explanation of this in the appendix may not have clearly articulated the motivation behind disabling Gemini’s safety filter. Our decision was driven by our threat model, which focuses on deliberate misuse by adversaries using the most capable API models. The safety filter in the Gemini API, while valuable, is optional and can easily be turned off. Therefore, we choose to disable the filter given that an adversary looking to misuse models would do the same. We recognize the importance of investigating the effectiveness of BoN jailbreaking with models defended by safety filters and consider it a promising direction for future research.
>
> > The study does not evaluate the audio quality of the modified samples post-attack, which is an important aspect for assessing the stealthiness of these attacks. [...] Without such an analysis, it is unclear if the BoN attacks are feasible in scenarios where high-quality audio is essential for covert operations.
>
> It is common in the adversarial robustness literature of speech-to-text models to care about the stealthiness of attacks. However, in our misuse threat model, we care about whether or not the ALM will ever elicit harmful information, even if the audio clip is easily detectable as adversarial by humans. LLMs are not safeguarded by humans who manually review the inputs, instead we rely solely on the alignment training or mandatory safeguards (unlike Gemini's filters) that LLM providers put in place. Our study aims to understand if a potential adversary can jailbreak these systems in a black-box manner by any means necessary. We do not claim nor expect BoN jailbreaking to be appropriate for covert operations -- it is a strong black-box algorithm for eliciting harmful information across any harmful request.
>
> > * How might the ASR be affected if the safety filter remained active? Including this would provide a clearer picture of BoN’s practical effectiveness and real-world implications. Would the authors consider conducting additional experiments with the safety filter enabled to better align the study with real-world conditions?
> > * Audio quality after attack modifications is an important factor for stealthy and practical attacks. Could the authors provide details on the perceptual quality of the modified audio samples, perhaps using a metric like the ViSQOL score?
>
> See our response to the related points above. We have updated the manuscript to motivate our choice to turn the safety filter off (see footnotes on page 3) and why stealthiness is not required given our misuse threat model (see thread model paragraph on page 4).
>
>
> > * The term "ASR" is commonly used to denote "Automatic Speech Recognition" in the speech research community. Could the authors consider using an alternative term?
>
> We agree with the reviewer that this is an unfortunate duplication of a common acronym in the speech recognition field. For this reason, we are careful to use speech-to-text instead of automatic speech recognition throughout the paper. In the adversarial robustness field, ASR for attack success rate is also commonly used. For this reason, we choose to continue to use ASR for attack success rate, but we thank the reviewer for raising the concern.
>
> We hope our response has addressed your feedback and that you consider strengthening your support for our paper. We welcome further discussion.

---

> ### Comment · Reviewer_YqsM · 2024-11-25
>
> Thanks for the clarification. But my concerns are not addressed by authors' rebuttal, I choose to keep the original score.

---

> > ### Author Response · Authors · 2024-11-25
> >
> > Could you please elaborate on which aspects of the concerns you feel were not adequately addressed, given the misuse threat model we care about? We aim to fully understand and consider your viewpoints.

---

### Author Response · Authors · 2024-11-25

We sincerely appreciate the detailed feedback provided by the reviewers on our submission. The positive remarks from Reviewers YqsM, t8Hb, and g1y2 highlight the valuable insights into the scalability of our Best-of-N (BoN) Jailbreaking algorithm and its well-structured presentation. We are encouraged by the acknowledgment of the novelty in exploring jailbreaking within the audio domain—an area that remains underexplored. Additionally, the appreciation for our comprehensive and complete experiments on audio language models (ALMs) from Reviewer g1y2 underscores the thoroughness and relevance of our research.

However, we acknowledge that there were several concerns regarding the methodology’s clarity, the practicality of our experiments, and the specificity of our findings to the audio domain. These concerns are insightful, and we have addressed each with revisions to our paper. It seems there may have been some misunderstanding of our motivations and the threat model due to possibly different backgrounds of the reviewers, particularly those more versed in automatic speech recognition rather than adversarial robustness. We hope our responses have clarified our choices and we are confident that the revised version will comprehensively address the reviewers’ feedback.

---

### Meta-Review · Area_Chair_mKbu · 2024-12-19

**Metareview:**

This work proposed a jailbreaking attack against Audio Language Models (ALMs). It received 3 detailed reviews. Most reviewers recognized that the studied task is interesting, and the proposed method is tailored for audio model.

However, there are several important concerns from reviewers, mainly including:
1. The threat model that the adversary can turn off the Gemini’s safety filter is not practical.
2. The stealthiness of the attacked sample is not studied.
3. The motivation that the naive attack against ALMs is poor is not well analyzed.
4. The writing is unclear that the proposed method is not clearly described.

There are sufficient discussions between authors and reviewers. The reviewers considered that the main concerns mentioned above were not well addressed. I agree with that comment, especially that the threat model of turning off ALM's safety filter is unreasonable.
Thus, my recommendation is reject. Hope these comments could help authors further improve this work.

**Additional Comments On Reviewer Discussion:**

As said above, the main concerns are not well addressed in the discussion.

---

### Decision · Program_Chairs · 2025-01-22

Reject